# Responsive Neurostimulation of the Anterior Thalamic Nuclei in Refractory Genetic Generalized Epilepsy: A Case Series

**DOI:** 10.3390/brainsci13020324

**Published:** 2023-02-14

**Authors:** Carly M. O’Donnell, Sara J. Swanson, Chad E. Carlson, Manoj Raghavan, Peter A. Pahapill, Christopher Todd Anderson

**Affiliations:** 1Department of Neurology, Medical College of Wisconsin, 8701 Watertown Plank Road, Milwaukee, WI 53226, USA; 2Department of Neurosurgery, Medical College of Wisconsin, 8701 Watertown Plank Road, Milwaukee, WI 53226, USA

**Keywords:** genetic generalized epilepsy (GGE), drug-resistant epilepsy, anterior thalamic nuclei (ANT), responsive neurostimulation (RNS), electrocorticography (ECoG), neuropsychological outcomes

## Abstract

Genetic generalized epilepsies (GGEs) are thought to represent disorders of thalamocortical networks. There are currently no well-established non-pharmacologic treatment options for patients with drug-resistant GGE. NeuroPace’s Responsive Neurostimulation (RNS) System was approved by the United States Food and Drug Administration to treat focal seizures with up to two ictal foci. We report on three adults with drug-resistant GGE who were treated with thalamic RNS. Given the severity of their epilepsies and the potential ictogenic role of the thalamus in the pathophysiology of GGE, the RNS System was palliatively implanted with leads in the bilateral anterior thalamic nuclei (ANT) of these patients. The ANT was selected because it was demonstrated to be a safe target. We retrospectively evaluated metrics including seizure frequency over 18–32 months. One patient required explantation due to infection. The other two patients were clinical responders. By the end of the observation period reported here, one patient was seizure-free for over 9 months. All three self-reported an improved quality of life. The clinical response observed in these patients provides ‘proof-of-principle’ that GGE may be treatable with responsive thalamic stimulation. Our results support proceeding to a larger study investigating the efficacy and safety of thalamic RNS in drug-resistant GGE.

## 1. Introduction

Genetic generalized epilepsy (GGE) is characterized by seizure activity with bilateral, synchronous onsets and no established focal brain region that drives its pathophysiology. Previously referred to as ‘idiopathic generalized epilepsy’, GGE is currently considered the preferred term by the International League Against Epilepsy (ILAE) [1,2]. While antiseizure medications (ASMs) are the primary treatment option for patients with GGE and are effective for many, over 20% are drug-resistant and have significantly increased risks of poor outcomes, including injury, lifestyle limitations, reproductive challenges, and sudden unexpected death in epilepsy [3]. Drug resistance is defined by the ILAE as a lack of sufficient clinical response to two or more tolerated and appropriate ASMs [4]. The etiology for drug resistance in this subset of GGE patients is unknown but is an area of active interest within the field [5].

Bilateral deep brain stimulation (DBS) of the anterior nuclei of the thalamus (ANT) via the DBS System (Medtronic) is approved by the United States Food and Drug Administration (FDA) as a treatment for medically drug-resistant focal onset seizures [6]. It has also been implanted off-label in the centromedian nucleus (CM) of the thalamus to treat GGE [7]. Unfortunately, side effects including neuropsychological changes and sleep disruption related to chronic stimulation of the thalamus are a potential concern, especially for those with preexisting psychiatric conditions [8,9,10,11].

The Responsive Neurostimulation (RNS) System by NeuroPace (Mountain View, CA, USA) is FDA approved to treat drug-resistant epilepsy and is associated with a 75% median seizure reduction in long-term follow-up [12]. Unlike the Medtronic DBS System, which provides fixed, cyclic stimulation to the ANT, the RNS System delivers stimulation only in response to epileptiform activity which it analyzes and records in real time. RNS patients thus receive less total stimulation per day than those with ANT-DBS, and the stimulation that they do receive is also more temporally targeted. These patients may consequently experience fewer adverse effects [13].

Although only approved to treat epilepsy with up to two ictal onset foci, the RNS System has been successfully implanted in the ANT of multifocal epilepsy patients with overall positive outcomes [14]. Furthermore, ANT-RNS was previously reported in a single GGE patient at Massachusetts General Hospital [15]. In this case, strips were placed in the bilateral posterior frontal cortex and depth electrodes were placed in the bilateral ANT. However, only the right posterior frontal cortex and ANT contacts were connected to the RNS device, as the patient’s seizures demonstrated asymmetric thalamocortical propagation with left hemibody predominance. As such, this GGE patient ultimately received unilateral thalamic stimulation. Bilateral ANT-RNS stimulation in GGE has not yet been reported.

The precise mechanisms that lead to the generation of seizures in patients with GGE remain unclear. Many studies have suggested that ictogenesis in GGE involves the thalamus and its closely related networks, even though epileptiform activity appears diffuse on scalp EEG [16,17,18,19]. Other hypotheses suggest a cortical focus or foci with rapid spread of electrical excitation (potentially via the thalamus) such that it is detected as generalized in onset on EEG [20,21]. Such rapid generalization would make identifying a cortical focus for the purpose of surgical resection or RNS lead placement impossible with existing technology. Nonetheless, both theories support the notion that responsive thalamic stimulation may provide a potential means to disrupt ictal build-up, interrupt malfunctioning cortico-thalamocortical circuity, or elevate seizure thresholds through neuromodulatory effects in GGE [19]. 

Bearing this in mind, the RNS System was implanted as a palliative intervention with leads in the bilateral ANT of three adult patients with severely drug-resistant GGE, who had also failed prior trials of vagal nerve stimulation (VNS) therapy. Clinical vignettes for all three patients are provided in the following sections.

## 2. Materials and Methods

### 2.1. Case 1

A 30-year-old woman presented with generalized tonic-clonic convulsions (GTCs), absence seizures, and myoclonic jerks. She first began having seizures at 10 years of age. She had no family history of epilepsy. Scalp EEG showed 4–5 Hertz (Hz) generalized spike-and-wave discharges. An MRI of the brain with a dedicated epilepsy protocol was normal. Her history of seizures was complicated by co-morbid depression and conversion disorder with psychogenic non-epileptic seizures (PNES). Her PNES was characterized on video EEG by episodes of verbal unresponsiveness; intermittent, non-rhythmic tremulousness of one or both hands; grunting; and periodic hyperventilation. 

This patient was diagnosed with juvenile myoclonic epilepsy (JME), and her diagnosis was confirmed in the Froedtert Hospital epilepsy monitoring unit (EMU) by the capture of absence seizures and a generalized convulsion. Treatment with stimulation of the ANT utilizing either the DBS or RNS was offered as a palliative measure, since she had been failed by prior trials of 22 ASMs as well as VNS. The patient elected to pursue the RNS System for two reasons: (1) potentially fewer neurocognitive side effects with lower amounts of total stimulation per day (an average of <6 min per day with RNS [22] compared to 4 h with DBS [8]) and (2) the diagnostic benefit of ambulatory electrocorticography (ECoG) given her dual diagnoses of GGE and PNES. 

The patient’s epilepsy team decided to place depth leads in her bilateral ANT and one cortical strip on the left frontal convexity. This lead configuration was chosen because the cortical strip would provide well-characterized neocortical morphology to epileptiform activity if the thalamic signal during generalized discharges proved suboptimal for triggering responsive stimulation or was difficult to interpret clinically. Of the thalamic nuclei, the ANT was selected because of its established safety profile [8]. It has been a site of implantation for both DBS and RNS leads [14]. At the time of this intervention, the results from the ANT-RNS GGE patient at Massachusetts General Hospital (described in the introduction) had not yet been reported. Of patient 1′s thalamic leads, only the right side was connected to the RNS System so that a signal from the cortex was also available. The team decided to place the left ANT lead so that bilateral thalamic stimulation could potentially be explored in the future. Further description of the surgical approach is described in Section 2.4.

Three years after RNS implantation, the patient reported new onset, frequently occurring headaches with associated nausea and vomiting. These felt like “waves” in her head. Symptom onset was not associated with a change in stimulation settings, which had been consistent for 6 months prior to headache onset. To explore the relationship between her headaches and the device, stimulation was temporarily turned off during a planned inpatient admission. With this change, her headaches worsened rather than improved; stimulation was successfully reintroduced. The headaches resolved independently.

### 2.2. Case 2

A 25-year-old woman presented with GTCs, absence seizures, absences with eyelid myoclonia, and seizures with falls (not clearly atonic and without electrodecrement on scalp EEG). Her seizures began at 11 years of age. She had no family history of epilepsy. Scalp EEG showed 3–5 Hz diffuse spike-wave runs lasting up to 20 s with rare slow spike-wave complexes, and rare generalized paroxysmal fast activity. Her brain MRI with dedicated epilepsy protocol was normal. In Froedtert Hospital’s EMU, she had very frequent diffuse epileptiform activity along with multiple absence seizures and rare eyelid myoclonia associated with scalp epileptiform activity. She was diagnosed with GGE, and more specifically, epilepsy with eyelid myoclonia (EEM)—formerly known as Jeavons syndrome.

Although she became a successful professional in adulthood, this patient’s epilepsy became progressively more severe and ultimately derailed her burgeoning career. She had been tried on 10 ASMs and VNS previously with persistence of seizures. She had a history of anxiety, depression, and suicidal ideation. These had resolved two years prior to her eventual RNS surgery.

Based on the severely refractory nature of her epilepsy, DBS was discussed as a potential treatment option during a clinic appointment. However, the patient and her mother requested more information on the RNS System, which they had researched independently. Given her appropriate candidacy and the previously successful ANT-RNS implantation of patient 1, RNS was ultimately offered to patient 2 as a palliative treatment option, which she accepted.

The RNS System leads were placed in her bilateral ANT. The clinical epilepsy team decided that cortical strips were unnecessary because the thalamic lead data from patient 1 were clear, easily interpreted, and the thalamic lead stimulation appeared therapeutic.

### 2.3. Case 3

A 57-year-old woman presented with GTCs and myoclonus. She reported several cousins with epilepsy, but no immediate family members. Her scalp EEG showed 2–3 Hz diffuse spike-and-wave runs along with polyspike and wave runs lasting up to 8 s. Her brain MRI was normal, as were her physical and neurological examinations. She was diagnosed with JME. Although atypical, as her seizures began at 3 years of age, she met all other criteria for this diagnosis. A total of 14 ASMs were trialed, but her seizures persisted. VNS was also trialed but did not result in significant seizure improvement. In addition to her epilepsy, she had reported well-controlled comorbid bipolar 1 disorder, depression, and conversion disorder with PNES. Her PNES was captured on video EEG and was characterized by episodes of unresponsiveness, intermittent grimacing, a rushing sensation in her head, and tingling in her bilateral hands and feet.

The patient’s seizures worsened with age. She voiced significant discontentment with her seizure control. As a result of this and because of the prior successes of patients 1 and 2, RNS was offered as a palliative measure. Discussion about the device and alternative medical options were ongoing for 10 months prior to surgery. Leading up to and at the time of RNS implantation, patient 3 was prescribed clonazepam as a monotherapy (Table 1). The patient made the decision with her clinical epilepsy team to pursue monotherapy along with ongoing VNS treatment due to her previous side effects from other ASMs and her severe drug resistance.

Like patient 2, RNS leads were placed in the bilateral ANT with no cortical strips. Her device was explanted after 553 days due to infection. The details of this event are described further in Section 3.1.3.

### 2.4. Surgical Procedures

All electrodes were placed with the patient in the supine position under general anesthesia. Targeting of the ANT was performed with stereotactic 3T MRI following the principles of Jiltsova et al. [23]. Targets and frontal surgical approaches were fed through the Stealth System (Medtronic, Minneapolis, MN, USA), interfaced with the Vertek articulating arm. After the insertion of a long cannula that extended to the targets along the planned trajectories, bilateral RNS electrodes were placed. Subsequently, the intra-operative Medtronic O-Arm^TM^ (Medtronic, Minneapolis, MN, USA) was used to confirm appropriate lead placement. Leads were then anchored to their respective burr holes in the usual fashion. At this point, a craniotomy was performed to fit the RNS ferrule, receiver, and battery. The bilateral RNS leads were then attached to the device.

For patient 1 (see Section 2.1), prior to placement of the RNS ferrule, the anterior dura was opened, and a strip electrode was placed over the lateral left frontal lobe. The lateral left frontal strip electrode and right-sided ANT RNS electrode were then attached to the RNS system after being placed into the ferrule. 

Prior to closing, all systems were interrogated to ensure satisfactory ECoG recordings. The patients tolerated the procedures well and the perioperative periods were unremarkable.

### 2.5. RNS Detection Parameters

After RNS System implantation, the patients were instructed to use their device magnets to trigger ECoG storage whenever they experienced a clinical event. In all three patients, rhythmic delta activity (2–4 Hz) was identified in the thalamic leads coincident with events marked with magnet triggers (see Figure 1), and in the case of patient 1, also coincident with generalized spike bursts recorded from the left frontal strip leads. These readings corresponded to typical GGE scalp EEG findings that are characterized by frequently normal background activity superimposed with paroxysmal generalized 2.5–5.5 Hz spike-waves [1]. For this reason, all bandpass detections were tuned to the recorded delta frequency range.

The RNS System can perform detections using any two of its four recording channels. To maximize the detection specificity for the desired 2–4 Hz signal, the two channels are selected to be those where the amplitudes of the rhythmic delta activity on each side are consistently maximal. For example, in patient 2, these were the distal contacts for the left (Figure 1, LANT 2–1) and the middle electrodes for the right (Figure 1, RANT 2–3). The rhythmic delta activity was bilateral in the magnet-triggered ECoGs. Finally, to further improve specificity, AND logic was employed, such that Pattern A1 (left rhythmic delta) and Pattern A2 (right rhythmic delta) must be present concomitantly for detection to occur.

### 2.6. RNS Stimulation Parameters

Previous reports of ANT stimulation on a wide range of patient populations have been summarized with encouraging results [24]. In these studies, subjects generally received high-frequency stimulation (>100 Hz) with open-loop long bursts (typically 1 min on, 5 min off). As a result of this, the three patients reported here were all programmed to receive high-frequency stimulation (125 Hz) with responsive 5 s bursts delivered to the electrodes where the rhythmic delta ECoG activity was maximal. The charge density was initially set to 0.5 µC/cm^2^ and was gradually increased at subsequent visits until a clinical response (i.e., reduction in patient-reported seizures and in long episodes quantified by ECoG recordings) was obtained (range 1.0 to 3.0 µC/cm^2^). Simultaneous changes to ASMs and device settings were avoided. VNS therapy for all three patients persisted, but the settings were not changed during the observed period reported here. The patients reported no side effects, specifically denying paresthesia or headache during or after stimulations and stimulation changes.

‘Parameter optimization’ was identified as the point when the treating physician concluded that further setting adjustments were unlikely to yield additional benefit based on the patient’s response. In other words, these were the parameters under which patients appeared to have achieved or nearly achieved clinical seizure freedom, and the risks of further setting adjustments (i.e., a possible increase in seizure frequency) appeared to outweigh the potential benefits (i.e., further reducing epileptiform activity or eliminating clinical seizures entirely). This term is new to the field, and the process of determining when parameter optimization has been reached is still evolving. The final stimulation parameters for all patients are detailed in Table 2. An example of responsive stimulation is illustrated by Figure 2. 

### 2.7. Statistics

All metrics were analyzed using GraphPad Prism 9 (version 9.4.0) and Microsoft Excel (version 16.62).

Unpaired *t*-tests with Welch’s correction were performed within each subject to compare total daily event detections and long episodes (a surrogate for clinical seizures [25]) pre- and post-‘parameter optimization.’ Patient 3 did not reach ‘parameter optimization’ as defined by her clinical team and so was excluded from this analysis. After the initial RNS implantation procedure, the detection parameters for all patients were slowly adjusted over time until the desired settings were obtained, as discussed in Section 2.4. Due to this, our analysis only includes the final detection settings used for each patient. All stimulation parameter changes that occurred during this single detection epoch were included in the analysis. It does not include all data from the first day of RNS System implantation. The start of the epoch with the final detection settings is defined as day 0. ‘Pre-optimization’ data included all time points between day 0 to the day that the ideal stimulation parameters were selected (which differed for each patient)—the detection parameters were not changed. ‘Post-optimization’ data included the day after the optimal stimulation parameters were programmed to the end of the observation period (see Figure 3). No stimulation parameter changes were made in the ‘post-optimization’ epoch.

Paired *t*-tests with Welch’s correction were performed within each subject to compare the rate of epilepsy-related hospitalizations pre- and post-RNS implantation. Post-RNS implantation data included all epilepsy-related hospitalizations from the date of implantation to the end of the observation period. We used the same number of days as existed in the post-RNS follow-up period for each patient to calculate the pre-implantation hospitalization rate. For example, patient 1 had 986 days of follow-up in the observation period (Table 1). This was compared to the 986 days prior to her surgery to determine her rate of epilepsy-related hospitalizations pre-implantation.

The results from these described analyses are shown in Table 3 and Figure 3. Given the small sample size, no statistical comparisons were performed between subjects.

## 3. Results

### 3.1. Epilepsy Outcomes

Two out of our three GGE patients were robust, statistically significant clinical responders to the ANT-RNS System (Table 3, Figure 3).

Initially, for all three patients, the detection rates for the delta frequency detectors were in the range of 200 to 500 per day. This resulted in approximately 18 to 42 min of daily stimulation. However, over time, patients 1 and 2 experienced significant reductions in detection rates without changes in the detection settings. This resulted in only 3 to 5 min of daily stimulation being delivered to each of these patients.

#### 3.1.1. Patient 1

Prior to RNS System implantation, patient 1 had approximately one GTC per year and experienced daily absence seizures that were often noticeable to the people around her. Her last GTC occurred 9 months after RNS implantation. After that seizure, in the 23 months of additional follow-up reported here, she had no further GTCs and no apparent seizures with loss of awareness.

Additionally, pre-implantation, comorbid PNES caused patient 1 to be hospitalized repeatedly. Since receiving the ANT-RNS System, she or family members have been able to ask her epilepsy team remotely whether she has experienced a seizure or psychogenic event based on her ECoG data within minutes of the event. Due to this, she has avoided a significant number of what would have been medically unnecessary emergency medevac flights and subsequent hospitalizations (Table 3).

#### 3.1.2. Patient 2

Patient 2 had almost daily clinical seizures pre-implantation (for example, 25 in one month), which included seizures leading to serious injury and, frequently, hospitalization. At the end of the observed period reported here, she had been seizure-free for 9 months, and her rate of epilepsy-related hospitalizations since implantation had reduced significantly (Table 3). Her ASMs were being reduced with no measurable increases in epileptiform activity in preparation for a future planned pregnancy. 

#### 3.1.3. Patient 3

Patient 3 believes that she experienced improved seizure control, but this was not supported by her ECoG recordings (Table 3). ‘Parameter optimization’ was never achieved given the early removal of her device due to complications (Figure 3). She had a chronic, superficial infection at the neurostimulator surgical site due to wound picking, which likely explains her apparent clinical worsening and her increase in hospitalization rate post-implantation (though this is not statistically significant). Approximately 18 months post-implantation, she presented with signs of an acute infection around the neurostimulator site. The wound was drained, bacteria were cultured, and she completed an antibiotic course with improved symptoms. However, at her one-month follow-up appointment the patient reported purulent drainage from the washout site. She was strongly advised by the neurosurgery team that the device should be removed but she declined because she felt it had improved her quality of life. She instead opted to pursue an additional washout with continued antibiotics. Before the second washout could be completed, she was found in status epilepticus likely secondary to sepsis. Her device was removed emergently 553 days after it was initially implanted. The infection resolved and she has since fully recovered with no residual neurological deficits.

### 3.2. Neuropsychological Outcomes

Pre-RNS, patient 1 demonstrated low-average intellectual abilities and impairments in working memory, processing speed, and verbal fluency. Her memory abilities were variable with impaired verbal memory for stories and impaired design learning, but she had low-average abilities at verbal list learning and the recall of geometric designs. The patient underwent post-operative neuropsychological testing two and a half years following her RNS placement. Her intellectual scores were stable, although her processing speed had modestly declined. Her memory scores improved from low-average to average for list learning and from impaired to low-average for design learning. Her naming and visual spatial skills were stable. She continued to report moderate depressive symptomatology, but reported significant improvements in her quality of life [26], with reduced seizure worry and medication side effects, improved social functioning, and an overall improved quality of life. This is demonstrated in Table 4.

The pre-implantation neuropsychological testing for patient 2 revealed average to low-average intellectual functioning, borderline range short attention span, and impaired processing speed. Her verbal and perceptual intellectual scores were stable following RNS, and her processing speed improved significantly (more than two standard deviations) from impaired to low-average. Her memory scores were largely average to low-average with the exception of impaired design learning. Post-implantation, her story and design memory scores were stable, and both list and design learning over repeated trials improved. Object naming was stable from before to after RNS. Visual spatial skills (judging line angle orientation and facial discrimination) improved post-implantation. Her level of self-reported depression declined from mild to minimal post-RNS. Her quality of life improved by over three standard deviations for overall quality of life, with improvements reported in seizures worry, emotional well-being, energy, cognitive functioning, medication effects, and social functioning.

Neuropsychologically, patient 3 had average intellectual abilities, average to high-average memory scores, and normal scores for language and visual spatial skills before RNS implantation. After receiving the device, she self-reported improved quality of life. Formal post-implantation neuropsychological testing was not completed due to device removal. Post-device explant neuropsychological testing had yet to be performed at the time of our analysis which concluded 97 days after her RNS removal. During this period, she appeared unchanged from her baseline functional status when in clinic after her recovery, despite the complications she experienced. No focal neurological deficits were noted on exam, she remained fully oriented, and she continued to engage in her medical care. There were no changes in her ability to engage in her usual activities of daily living.

While the sample size is limited, pre- and post-RNS cognitive testing indicated stable intellectual and language skills in patients 1 and 2. Processing speed changes were variable. Memory and visual spatial skills were stable to improved. Quality of life was significantly improved. Patient 2, who saw the greatest benefit from RNS in terms of seizure control, also had an excellent outcome in terms of cognition, mood, and quality of life, with a dramatic improvement in her processing speed as well as improvements in memory and visual spatial skills, and an apparent improvement in QOL.

## 4. Discussion

To our knowledge, this is the first multi-subject series of GGE patients to have received ANT-RNS, and it is also the first time that bilateral ANT-RNS stimulation in GGE has been reported. The robust seizure reduction noted in patients 1 (unilateral thalamic stimulation) and 2 (bilateral thalamic stimulation) illustrates the potential utility of RNS for a broader range of epilepsies than focal syndromes alone.

The original goals for implanting these three drug-resistant GGE patients with the ANT-RNS System were palliative. Their epilepsies not only reduced their quality of life, but also regularly resulted in severe injuries and subsequent hospitalizations. With published safety data regarding thalamic RNS as well as ANT-DBS, evidence supporting thalamic focality in generalized epilepsies, and the severely refractory nature of their seizures, the treating team felt this intervention was appropriate for these three patients, although it was off-label. Ultimately, it was hoped that this novel treatment would result in improved seizure control and quality of life. This was clearly the case for two out of our three patients. Unfortunately, patient 3 did not reach parameter optimization as the device was removed due to infection which, in general, is the greatest risk associated with RNS [13]. Given our small sample size, an infection rate of 33.3% is not obviously discordant with those shown by other trials investigating thalamic neurostimulation devices [8,12].

Formal neuropsychological testing of the two patients who will continue with their RNS treatment showed unchanged or improved scores post-implantation. This runs parallel with the findings of Loring et al. observed during the RNS System clinical trials [22]. Quality of life, as assessed formally by the QOLI, also improved in our patients post-implantation. The issue of adverse neuropsychological changes, particularly mood and memory, potentially caused by thalamic stimulation, is a significant concern in the field of epilepsy [27]. Our results suggest that the reduced total daily stimulation associated with thalamic RNS may potentially mitigate the risk of adverse mood and cognitive effects as compared with the higher amount of stimulation associated with DBS, although the limited sample size precludes any definitive conclusions.

The ECoG amplitudes shown in Figure 1 and Figure 2 are markedly asymmetric in amplitude although previous continuous EEG in patient 2 showed gross left–right symmetry. Asymmetry on scalp EEG is common in GGE, with some reports indicating that this finding is present to at least a small degree in 40–60% of cases [28,29]. While it is possible that the asymmetry on the ECoG in Figure 1 and Figure 2 is due to the placement of the RNS leads, it seems more likely that this phenomenon is related to that which causes asymmetry on scalp EEG. Although no clear mechanism has yet been identified, this is an area of active interest.

The mechanisms by which RNS treatment reduces seizures is an area of continued study. Although the basis for RNS therapy was initially believed to be the ability of brief pulse stimulation to abort localized ictal buildup, continued reduction in seizures over time without further changes in stimulation or detection parameters in patients with focal epilepsy have suggested that neuromodulatory effects on the epileptogenic network may reduce the initiation of seizures over time [12,30]. In our two clinical responders, acute reductions in both daily event detections and long episodes suggest that acute abortive effects likely contribute to the response. However, total detected events—including those too short to trigger a treatment—continued to show progressive decreases over time despite unchanged detection parameters. This suggests that the ANT-RNS System also produced a beneficial neuromodulatory effect that reduced overall epileptiform burden over time.

These results are, of course, limited by our small sample size. We do not have sufficient statistical power to speak with certainty about the generalizable benefits of ANT-RNS for all GGE patients or GGE subtypes. Given that this analysis was performed retrospectively and that all patient interventions were provided for clinical purposes, potential confounding factors including medication changes were not controlled for. However, changes to ASMs were avoided in appointments when RNS settings were adjusted. Thus, the differences in epileptiform activity following RNS setting revisions reported here can be attributed to the effects of the system itself rather than to ASMs. Finally, the time intervals between implantation and neuropsychological testing varied between our patients which, along with the small sample size, limit our ability to make between-subject comparisons.

Despite these limitations, given the positive clinical outcomes and the lack of post-implantation neuropsychological deficits, our results support proceeding to a larger prospective trial investigating the safety and efficacy of thalamic RNS in GGE.

## Figures and Tables

**Figure 1 brainsci-13-00324-f001:**
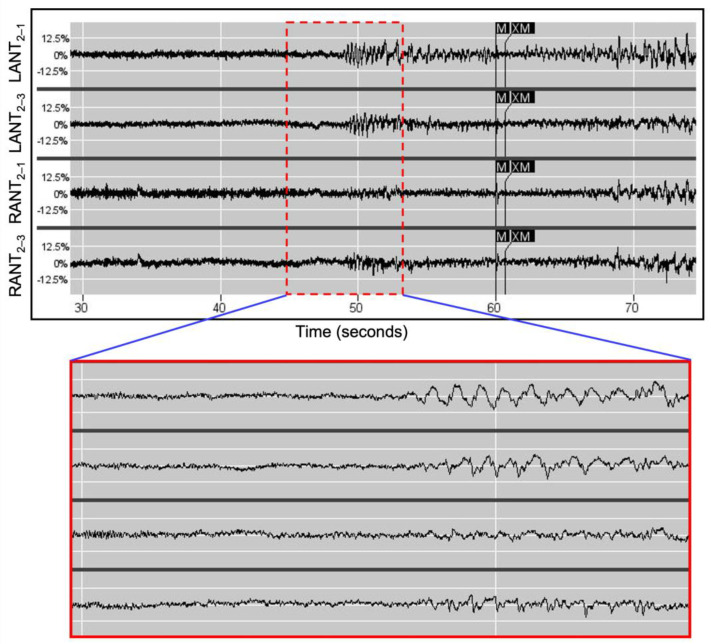
ECoG detection in patient 2 from NeuroPace’s Patient Data Management System (PDMS). The **top panel** is a segment from a 90 s magnet-triggered stored ECoG showing rhythmic delta activity (red dashed box) preceding magnet placement (M and XM). The **bottom panel** shows a magnified view of the ECoG where rhythmic delta occurs. The sensing montage for patient 2 was configured to omit electrode 4 (the most proximal) on both leads because a transventricular approach to the ANT was used for implantation. The impedance values from the bilateral, proximal electrodes were consistent with placements in the lateral ventricle (as these impedances are approximately half that of other electrodes).

**Figure 2 brainsci-13-00324-f002:**
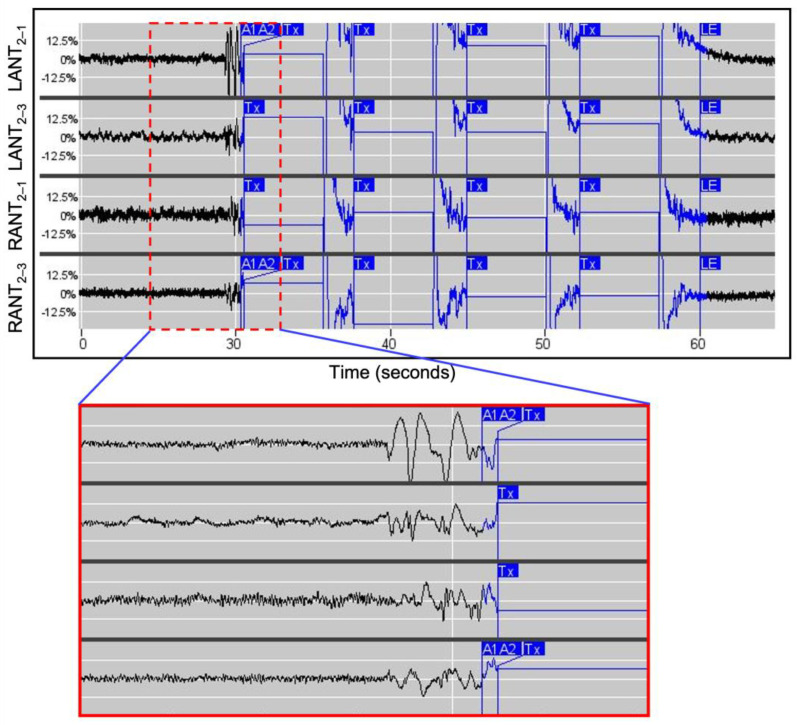
Stimulation in response to epileptiform activity, as shown in the PDMS. The **top panel** shows a detected event resulting in 4 out of 5 possible therapies (Tx). The therapy period is a flat blue line with a 5 s duration. The **bottom panel** shows a magnified view of the ECoG when seizure activity was initially detected, and the first treatment subsequently administered. After each therapy, the neurostimulator analyzes the ECoG and decides whether the next therapy should be delivered. When the ECoG trace changes from black to blue, this indicates that the neurostimulator has determined that its programmed detection criteria are being met. When the ECoG turns from blue to black, the neurostimulator has determined that the detection criteria are no longer being met, which ends the episode. These recordings are from patient 2.

**Figure 3 brainsci-13-00324-f003:**
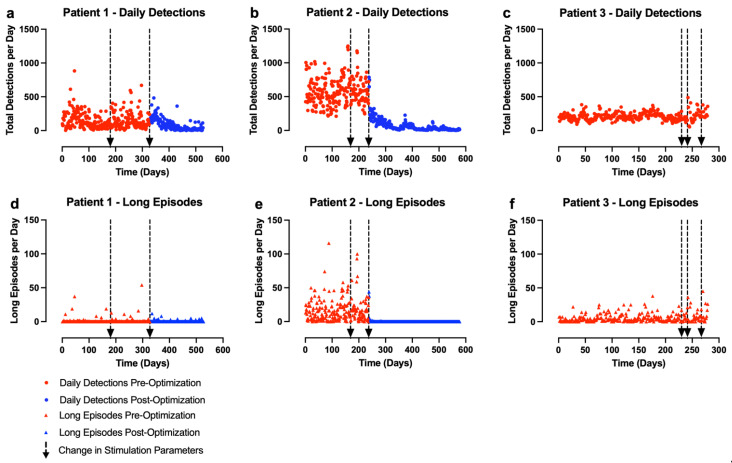
ANT-RNS treatment effects. Graphs depict number of epileptiform events (**a**–**c**) or long episodes (**d**–**f**) detected per day over time recorded on ECoG. Detection parameters remained constant throughout the above epochs. Day 0 is the first day of this detection epoch, not the first day of RNS implantation. Dashed arrows indicate when stimulation parameters were changed.

**Table 1 brainsci-13-00324-t001:** Clinical and demographic information.

Subject ID	Patient 1	Patient 2	Patient 3
Sex	Female	Female	Female
Age at onset (years)	10	11	3
Age at RNS implantation (years)	30	25	57
Epilepsy diagnosis and subtype	GGE-JME	GGE-EEM	GGE-JME
Psychiatric comorbidities	Depression, conversion disorder, history of suicidal ideation	Depression, anxiety, history of suicidal ideation	Depression, conversion disorder, bipolar 1 disorder
Comorbid PNES	Yes	No	Yes
Ongoing VNS pre- and post-RNS implantation	Yes	Yes	Yes
Number of ASMs trialed pre-implantation	22	10	14
Concurrent ASMs at RNS implant	CZP, DZP, LZP, TPM, medical marijuana	BRV, CZP, LEV, LPM, LZP, PB, ZNS	CZP
RNS electrode locations	LFC, LANT (turned off), RANT	LANT, RANT	LANT, RANT
Length of post-RNS implantation follow-up (days)	986	689	553
Length of post-RNS explantation follow-up (days)	NA	NA	97

ASMs: CZP = clonazepam, DZP = diazepam, LZP = lorazepam, TPM = topiramate, LPM = lacosamide, PB = phenobarbital, BRV = brivaracetam, ZNS = zonisamide, LEV = levetiracetam. RNS electrode locations: LFC = left frontal cortex, LANT = left ANT, RANT = right ANT. NA = not assessed.

**Table 2 brainsci-13-00324-t002:** Final RNS stimulation parameters.

Subject ID	Patient 1	Patient 2	Patient 3
Charge density	1.0 µC/cm^2^	3.0 µC/cm^2^	3.0 µC/cm^2^
**Burst 1**			
Electrode location(s)	LFC	LANT & RANT	LANT & RANT
Frequency	200 Hz	125 Hz	200 Hz
Current	2.0 mA	3.0 mA	3.0 mA
Duration	100 ms	5000 ms	5000 ms
**Burst 2 ***			
Electrode location	RANT	NA	NA
Frequency	125 Hz	NA	NA
Current	0.5 mA	NA	NA
Duration	5000 ms	NA	NA

* Patient 1 received two asynchronous bursts of stimulation, whereas the other two patients were programmed to receive simultaneous stimulation to both thalamic leads.

**Table 3 brainsci-13-00324-t003:** ANT-RNS treatment effects.

Subject ID	Patient 1	Patient 2	Patient 3
Daily Detections * Post-‘Parameter Optimization’	45% reduction (*p* = < 0.0001) ^†^	89% reduction (*p* = < 0.0001) ^†^	NA
Long Episodes Post-‘Parameter Optimization’	61% reduction (*p* = 0.0396) ^†^	98% reduction (*p* = < 0.0001) ^†^	NA
Epilepsy-Related Hospitalization Rate Post-RNS Implantation	75% reduction (*p* = 0.0142) ^†^	75% reduction (*p* = 0.0005) ^†^	200% increase (*p* = < 0.1575)

* Daily event detections by the RNS System; ^†^ results are statistically significant (*p* < 0.05).

**Table 4 brainsci-13-00324-t004:** Neuropsychological outcomes pre- and post-RNS implantation.

Subject ID	Patient 1 Pre-RNS	Patient 1 Post-RNS	Patient 2 Pre-RNS	Patient 2 Post-RNS	Patient 3 Pre-RNS	Patient 3 Post-RNS
Days before/after RNS implantation	772	912	130	753	685	NA
**IQ Standard Scores**						
VCI	85	87	91	96	95	NA
PRI	88	90	81	86	94	NA
WMI	70	70	77	74	92	NA
PSI	68	59	62	89	94	NA
**Percentiles**						
Story Memory	<1st	NA	25th	16th	63rd	NA
Design Memory	9th	NA	50th	37th	75th	NA
List Learning	16th	30th	8th	23rd	25th	NA
Design Learning	<1st	16th	1st	16th	NA	NA
Object Naming	9th	9th	10th	10th	91st	NA
Judging Line Angles	56th	72nd	4th	40th	56th	NA
Face Discrimination	85th	NA	90th	>98th	NA	NA
Depression (BDI)	Moderate	Moderate	Mild	Minimal	Severe	NA
T-Score QOL	T = 22	T = 36	T = 26	T = 61	T = 26	NA
Executive Functioning (Trails B)	12th	1st	<1st	5th	NA	NA

VCI = Verbal Comprehension Index, PRI = Perceptual Reasoning Index, WMI = Working Memory Index, PSI = Processing Speed Index—all from the Wechsler Adult Intelligence Scale-IV; BDI = Beck Depression Inventory; QOL = Quality of Life in Epilepsy (QOLIE-31).

## Data Availability

The data presented in this study are available on request from the corresponding author. The data are not publicly available in order to protect the privacy of our patients.

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
