# Peer review of "Responsive Neurostimulation of the Anterior Thalamic Nuclei in Refractory Genetic Generalized Epilepsy: A Case Series"

_brainsci, 2023, doi:10.3390/brainsci13020324_

Round 1
Reviewer 1 Report
The authors report 3 patients who underwent RNS implantation for intractable genetic generalized epilepsy (GGE). RNS was effective in 2 patients for seizures and cognitive function, while one patient had to decline the treatment due to complications.
This is an informative report, which suggests the usefulness of RNS for generalized epilepsy. Here are some minor comments.
1) Since the authors mention 'Genetic' generalized epilepsy, it would be informative to present the results of genetic testing (if any).
2) The authors introduce two hypotheses of the mechanisms of GGE, one of which mainly involves on thalamus, and a possible cortical focus in another. It might be interesting to discuss the possibility of applying RNS for a certain 'cortical focus'.
3) In Table 2, two columns indicate 'Patient 2'.
4) In Line 374, 'Readers of this manuscript likely noticed that' should be removed.
Author Response
Thank you for your comments. Please see the attached document for our detailed responses.

Reviewer 2 Report
Interesting article, but a small number of cases, actually only 2 patients can be assessed. You cannot write that in 3 patients there was an improvement in the quality of life, because in the third patient there were major complications. Your patient's psychological status should be assessed separately due to significant adverse events. In addition, this patient was treated with only one drug. Why, after treatment with one drug, an operative form of treatment was chosen, which caused significant side effects. Where is the consent of the bioethics committee. This patient should be described in detail separately, because according to the data from the article, he did not meet the criterion for drug-resistant epilepsy - no other drugs. Why was he treated only with clonazepam. The work requires a lot of explanation on this topic.
Author Response

(The authors gave the same response as above.)

Reviewer 3 Report
Dear Editors, Dear Authors,
the manuscript is an effective description of a small but significant case series.
I have no requests for changes or additions.
It is advisable to check that there are no typing errors (example: Table 2, "Patient 2" appears twice in the header).
Please see the attachment for more detailed comments.
With king regards.

Author Response

(The authors gave the same response as above.)

Reviewer 4 Report
The poor outcome in refractory GGE is well known; very frequent seizures, injuries, drugs adverse events and SUDEP can occurre in up to 20% of these patients.
Surgical approaches like VNS and DBS can result ineffective or burdened by neuropsicological changes and sleep disruption. The RNS in the ANT, although off-label, represents a very interesting new appoach. The GGE phisiopathogenesis remain unclear: the double hypothesis of a thalamic ictogenesis or of a very rapid cortico-thalamic diffusion, are not in contrast with the possibility that the thalamic electrical stimulation could play a neuromodulatory effect in these epilepsies. The marked reduction of the current amount delivered, morever, determining fewer adverse effects.
The study is well designed and performed. The weakness is the small number of the subjects : 2 patients, (while the third one confirms the high percentage of infections in RNS). Nevertheless the clear cut of the results bear up to continue with this appoach, having in mind to confirm both the clinical results as well as to elucidate the pathogenesis of GGE.
The method reported are completely missing of any surgical data. Not the approach, nor the control of the electrodes localization , neither by imaging nor by neurophisology. While the clinical and neurophisiological results are well documented, the process and the choice of the final stimulation parameters are not indicated. The 'optimization of the parameters' remains unclear as well as the patients feelings during the stimulation.
In agreement with the AA conclusion, these results support proceeding to a larger prospective trial investigating the safety and efficacy of ANT RNS in GGE
Author Response

(The authors gave the same response as above.)

Reviewer 5 Report
This is an interesting study on three adults with drug-resistant genetic generalized epilepsies (GGE) who were treated with responsive neurostimulation (RNS) in the bilateral anterior thalamic nuclei (ANT). Two of these patients were clinical responders, one was seizure-free for over 9 months. So RNS is proposed as a palliative treatment for patients with drug-resistant GGE.
Clinical and neurophysiological data are clearly reported and discussed.
As a minor point, I would ask to the Authors if they recorded a (video)EEG of the psychogenic non-epileptic seizures (PNES) in patient 1 and 3, or if PNES were diagnosed only anamnestically.
Author Response
Thank you for your comments. Please see the attached document for our detailed response.

Round 2
Reviewer 2 Report
I am satisfied with the explanations
Reviewer 4 Report
Your answers to my comments are satisfying.The metodological description for a working progress procedure usefull.
I hope that you are continuing with this new and very promising treatment both for the clinical results and for the study of the thalamic role in epileptogenesis.